# Integrated Analysis of Tissue-Specific Gene Expression in Diabetes by Tensor Decomposition Can Identify Possible Associated Diseases

**DOI:** 10.3390/genes13061097

**Published:** 2022-06-20

**Authors:** Y-H. Taguchi, Turki Turki

**Affiliations:** 1Department of Physics, Chuo University, Tokyo 112-8551, Japan; 2Department of Computer Science, King Abdulaziz University, Jeddah 21589, Saudi Arabia; tturki@kau.edu.sa

**Keywords:** gene expression, tensor decomposition, diabetes mellitus, neurodegenerative diseases

## Abstract

In the field of gene expression analysis, methods of integrating multiple gene expression profiles are still being developed and the existing methods have scope for improvement. The previously proposed tensor decomposition-based unsupervised feature extraction method was improved by introducing standard deviation optimization. The improved method was applied to perform an integrated analysis of three tissue-specific gene expression profiles (namely, adipose, muscle, and liver) for diabetes mellitus, and the results showed that it can detect diseases that are associated with diabetes (e.g., neurodegenerative diseases) but that cannot be predicted by individual tissue expression analyses using state-of-the-art methods. Although the selected genes differed from those identified by the individual tissue analyses, the selected genes are known to be expressed in all three tissues. Thus, compared with individual tissue analyses, an integrated analysis can provide more in-depth data and identify additional factors, namely, the association with other diseases.

## 1. Introduction

Gene expression analysis is an important step for investigating diseases and identifying genes that can be used as therapeutic targets or biomarkers or genes that are causes of disease. Although the development of high throughput sequencing technology (HST) has led to continuous increases in the amount of gene expression profile data, methods of integrating multiple gene expression profiles are still being developed. Tensor decomposition (TD) is a promising candidate method for integrating multiple gene expression profiles. Using this method, gene expression profiles from multiple tissues of individuals can be stored as a tensor $x_{ijk}\in R^{N\times M\times K}$, which represents the gene expression of the *i*th gene in the *j*th individual of the *k*th tissue. TD provides a method of decomposing a tensor into a series expansion of the product of singular value vectors, each of which represents a gene assigned to a specific individual or tissue. For example, by applying the higher-order singular value decomposition (HOSVD) method to $x_{ijk}$, we can obtain the following:(1)$$x_{ijk}=\sum_{\ell_{1}=1}^{N}\sum_{\ell_{2}=1}^{M}\sum_{\ell_{3}=1}^{K}G\left(\ell_{1}\ell_{2}\ell_{3}\right)u_{\ell_{1}i}u_{\ell_{2}j}u_{\ell 3k}$$
where $G\in R^{N\times M\times K}$ is a core tensor, $u_{\ell_{1}i}\in R^{N\times N},u_{\ell_{2}j}\in R^{M\times M},u_{\ell 3k}\in R^{M\times M}$ are singular value matrices and orthogonal matrices. We previously proposed a TD-based unsupervised feature extraction (FE) method [1] and applied it to a wide range of genomic sciences. Recently, this method was improved by the introduction of standard deviation (SD) optimization and applied to gene expression [2], DNA methylation [3], and histone modification analyses [4]. Nevertheless, because the updated method was only previously applied to gene expression measured by HST, whether it is also applicable to gene expression profiles retrieved by microarray technology remains to be clarified. In this paper, an integrated analysis was performed by applying the recently proposed TD-based unsupervised FE method with SD optimization to microarray-measured gene expression data for diabetes mellitus from multiple tissues. We found that applying the TD-based unsupervised FE with SD optimization to gene expression profiles from individual tissues can identify diseases associated with diabetes that cannot be identified by the other state-of-the-art methods.

There are multiple benefits to using TD to identify DEGs. First, since it is not a supervised method, it can select DEGs that are biologically more plausible than those selected using supervised methods. This can be explained using the following example wherein the aim is to identify DEGs that are distinct between two classes, e.g., patients and healthy controls. Supervised methods attempt to identify DEGs associated with a smaller divergence within individual classes, whereas TD allows one to select DEGs with within-class divergence to some extent (since TD tries to identify the representative state of distinction between two classes). If the representative state is associated with within-class divergence that has biological origins, e.g., age and sex, this divergence should not be penalized. However, supervised methods often do so, whereas the unsupervised method allows biological within-class divergence. Second, TD can select more stable DEGs, i.e., those independent of specific sets of samples considered in the analysis. This is because TD attempts to identify DEGs coincident with those of the representative state, which should be robust. Since sub-sampling does not change the representative state drastically, the gene set selected by TD is not altered drastically either. Third, TD can deal with multiple conditions. For example, if gene expression is measured in various tissues of several people, it is natural to format them as gene × person × tissue, which results in a tensor form. We have listed only a few important advantages here. Readers interested in acquiring information on other advantages of TD can refer to our recent book [1].

## 2. Materials and Methods

### 2.1. Gene Expression

Gene expression profiles (GSE13268, GSE13269, and GSE13270 [5]) were retrieved from the Gene Expression Omnibus (GEO), and they were obtained from a study of the progression of diabetes biomarker diseases in the rat liver, gastrocnemius muscle, and adipose tissue. Each of these profiles is composed of gene expression profiles from five individuals seen in two strains, Goto-Kakizaki and WistarKyoto, and they include data for three tissues (adipose, muscle, and liver) obtained at five time points after treatment. Three files named GSE13268_series_matrix.txt.gz, GSE13269_series_matrix.txt.gz, and GSE13270_series_matrix.txt.gz were downloaded from the Appendix A in GEO.

Gene expression profiles were formatted as a tensor, with $x_{ijkmst}\in R^{31099\times 5\times 5\times 2\times 2\times 3}$, representing the expression of the *i*th probe in the *t*th tissue (t=1: adipose, t=2: muscle, t=3: liver) at the *j*th time point for the *k*th replicate and *m*th treatment at the *s*th strain. These values are normalized as follows:(2)$$\begin{array}{ccc} \sum_{i}x_{ijkmst} & = & 0 \end{array}$$(3)$$\begin{array}{ccc} \sum_{i}x_{ijkmst}^{2} & = & 31099 \end{array}$$

### 2.2. Methods

Figure 1 shows the analysis pipeline. Methodological details can be found in the Appendix A.

## 3. Results

To validate the selected genes, 2281 gene symbols are uploaded to Enrichr [6] (For the full list of selected probes, genes, and enrichment analyses, check the Appendix A). Table 1 shows the results of the “KEGG 2021 Human” category in Enrichr. Since none of the terms are related to diabetes except for the top term, i.e., “diabetic cardiomyopathy”, the process initially appears to be a failure. Nevertheless, a number of the identified diseases are deeply related to diabetes mellitus. For example, many neurodegenerative diseases are listed, and diabetes mellitus is widely known to be a risk factor for neurodegenerative diseases [7,8,9,10,11]. Moreover, diabetes mellitus is known to be associated with thermogenesis [12], oxidative phosphorylation [13], and the PPAR signaling pathway [14]. Thus, the proposed method is successful in contrast to the first impression and can identify many diseases associated with diabetes mellitus.

Table 2 shows the top 10 terms in the category “ARCHS4 tissues” in Enrichr. Remarkably, gene expression is measured for three of the top four tissues. Similar results are found for the “Mouse Gene Atlas” category in Enrichr (Table 3). In conclusion, the proposed method is successful.

## 4. Discussion

Although the proposed method successfully integrated gene expression data measured in three tissues and identified diseases associated with diabetes mellitus, the identified genes also included genes expressed in all three tissues. If other methods that do not require an integrated analysis can perform similarly, then complicated methods, such as the proposed method, will not be required. To determine whether methods without integration can achieve similar performance, we tested three methods: *t* test, SAM [15], and limma [16]. Since the *t* test and SAM methods cannot simultaneously consider the distinction between the control and treatment as well as the dependence on time, we attempted to identify genes that presented expression differences between the control and treatment (no consideration of time dependence). For more details on how to perform these three methods, check the sample R source code in the Appendix A.

Table 4 shows the number of probes selected by the other methods. These methods select fewer probes than the proposed method (2542 probes), and the number selected in muscle is relatively low. According to the limma method, only two probes could be selected for muscle; thus, the method was not successful. The integrated analysis likely helped identify more probes, which resulted in more significant enrichment.

To further validate the genes selected by other methods, we converted probe IDs to gene symbols and uploaded them to Enrichr. Table 5 presents the results for the other methods on the “Mouse Gene Atlas” category in Enrichr. For muscle, neither SAM nor *t* test could select muscle as top ranked tissues whereas limma could identify only two probes as muscle-specific genes (see Table 4). Thus, the other methods are not better than the proposed method which could identify muscle specificity correctly (Table 3). Figure 2 shows the Venn diagrams between selected genes. Since the proposed method selects different genes from those specifically selected in individual tissues, an integrated analysis is a valuable method.

Finally, based on the genes associated with probes shown in Table 4, we found that the “KEGG 2021 Human” category in Enrichr does not include neurodegenerative diseases (see the Appendix A). Thus, the association between neurodegenerative diseases and diabetes mellitus can be found only when an integrated analysis, such as the proposed method, is employed. In this sense, an integrated analysis is more than a simple union of individual analyses and can identify factors that cannot be identified by individual analyses, such as potentially associated diseases. Thus, an integrated analysis of gene expression profiles in individual tissues provides more in-depth information than individual analyses, at least for certain cases. Thus, integrated analyses of gene expression profiles in individual tissues should be encouraged.

It may be plausible for other integrated methods to perform similarly. If this is true, the advanced methods that we have proposed here are not required. To rule out this possibility, we apply ComBat [17] to remove the batch effect between the three tissue types since we selected genes whose expressions are independent of tissues as can be seen in Appendix A; Table 4 shows the results. It is seldom reported to be successful. Limma failed to select any DEGs, and the numbers of genes selected by the *t* test and SAM are markedly different from each other in contrast to the identification of tissue-specific DEGs, whose numbers are more coincident across the three methods (Table 4).

Biological validation is also worse; Table 5 shows the result of the “Mouse Gene Atlas”. None of the tissues used in the experiments are listed, whereas the proposed method is (Table 3). In addition to this, based on the genes associated with probes shown in Table 4, we found that the “KEGG 2021 Human” category in Enrichr does not include neurodegenerative diseases (see the Appendix A) that were detected using the proposed method (Table 1). In conclusion, integrated analysis using ComBat is inferior to the proposed method.

One might wonder why an integrated analysis of three tissues from patients with diabetes mellitus can identify associations with neurodegenerative diseases. The PCA and TD-based unsupervised FE methods are frequently able to detect disease associations. We previously identified an association between cancer and amyotrophic lateral sclerosis [18] without investigating cancer gene expression and an association between heart diseases and posttraumatic stress disorder [19] without investigating brain gene expression. Therefore, we were not surprised that the integrated analysis using the proposed method was able to identify disease associations. To our knowledge, few studies have attempted to predict the association between diseases using gene expression, although many studies have focused on the associations between genes and disease [20,21,22] and between drugs and disease association [23,24,25]. Our proposed strategy would be useful for such studies.

## 5. Conclusions

In this study, we applied the proposed TD-based unsupervised FE with SD optimization method to perform an integrated analysis of gene expression measured in three distinct tissues using microarray architecture; moreover, the proposed method had not been applied to such data in previous studies. The results show that the proposed method can identify more genes than individual analyses. The selected genes are known to be expressed in all three tissues, and they are also enriched in many neurodegenerative diseases that have a known association with diabetes mellitus but cannot be identified by individual analysis. In this sense, integrated analyses might have the ability to identify additional factors relative to individual analyses.

## Figures and Tables

**Figure 1 genes-13-01097-f001:**
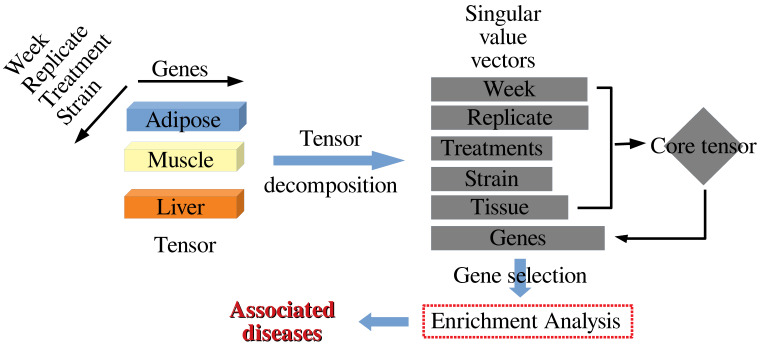
Overall flowchart of the analysis pipeline.

**Figure 2 genes-13-01097-f002:**
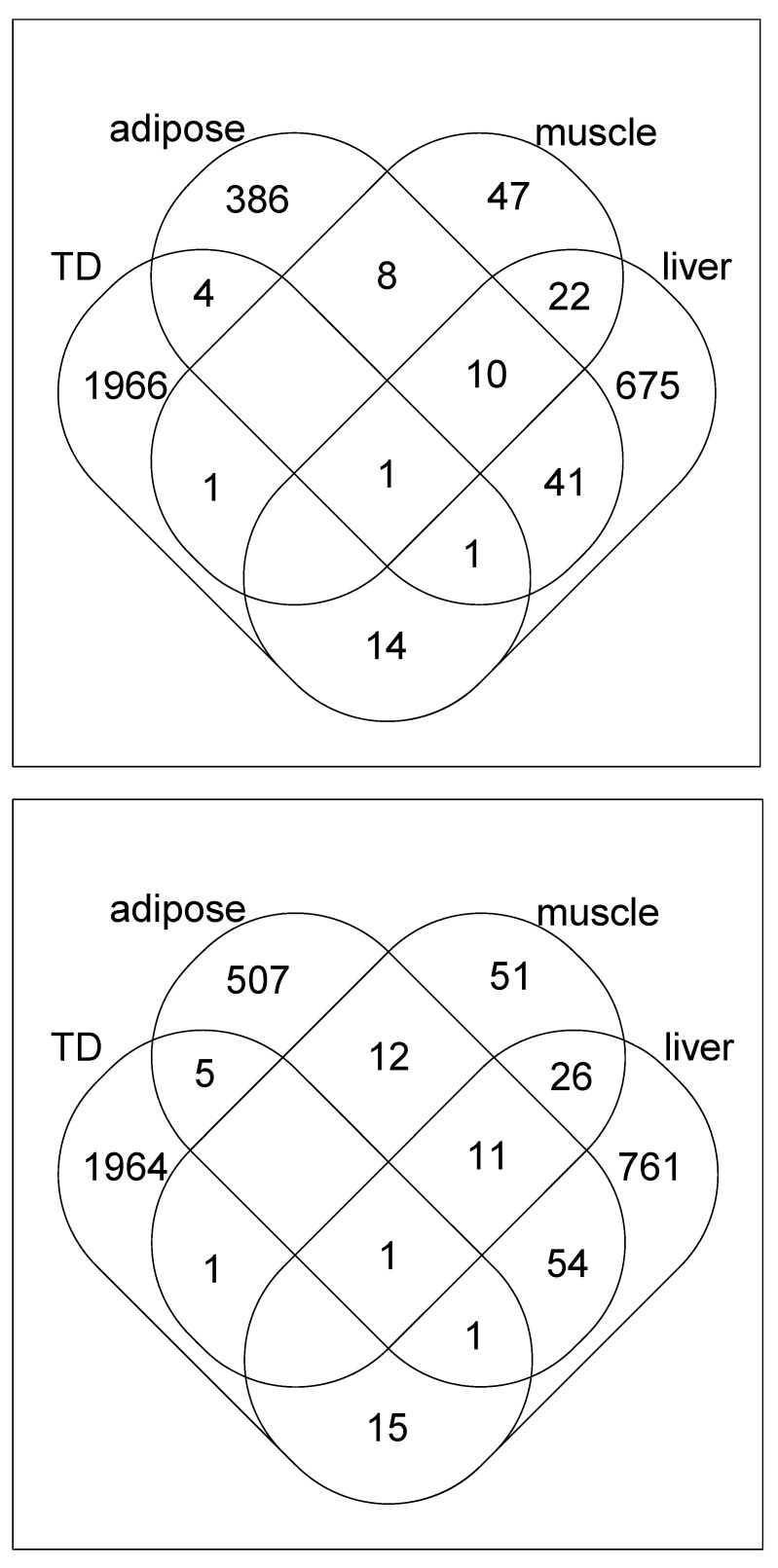
Venn diagrams between genes selected by various methods. Upper: *t* test, lower: SAM.

**Table 1 genes-13-01097-t001:** Top 10 “KEGG 2021 Human” category terms in Enrichr.

Term	Overlap	*p*-Value	Adjusted *p*-Value
Diabetic cardiomyopathy	83/203	$1.89\times 10^{-31}$	$5.80\times 10^{-29}$
Prion disease	93/273	$7.40\times 10^{-28}$	$1.13\times 10^{-25}$
Parkinson disease	86/249	$2.50\times 10^{-26}$	$2.55\times 10^{-24}$
Oxidative phosphorylation	60/133	$7.92\times 10^{-26}$	$6.06\times 10^{-24}$
Nonalcoholic fatty liver disease	65/155	$1.19\times 10^{-25}$	$7.30\times 10^{-24}$
Thermogenesis	76/232	$8.28\times 10^{-22}$	$4.22\times 10^{-20}$
Complement and coagulation cascades	42/85	$2.26\times 10^{-20}$	$9.85\times 10^{-19}$
PPAR signaling pathway	39/74	$2.58\times 10^{-20}$	$9.85\times 10^{-19}$
Alzheimer disease	94/369	$3.99\times 10^{-18}$	$1.36\times 10^{-16}$
Huntington disease	83/306	$6.48\times 10^{-18}$	$1.98\times 10^{-16}$

**Table 2 genes-13-01097-t002:** Top 10 terms in the “ARCHS4 Tissues” category in Enrichr.

Term	Overlap	*p*-Value	Adjusted *p*-Value
LIVER (BULK TISSUE)	481/2316	$3.49\times 10^{-63}$	$3.77\times 10^{-61}$
VENTRICLE	449/2316	$1.67\times 10^{-49}$	$9.04\times 10^{-48}$
SKELETAL MUSCLE (BULK TISSUE)	428/2316	$2.34\times 10^{-41}$	$8.42\times 10^{-40}$
ADIPOSE (BULK TISSUE)	410/2316	$6.46\times 10^{-35}$	$1.75\times 10^{-33}$
MYOBLAST	409/2316	$1.42\times 10^{-34}$	$3.08\times 10^{-33}$
SUBCUTANEOUS ADIPOSE TISSUE	401/2316	$6.92\times 10^{-32}$	$1.25\times 10^{-30}$
ATRIUM	366/2316	$2.38\times 10^{-21}$	$3.67\times 10^{-20}$
HEART (BULK TISSUE)	363/2316	$1.53\times 10^{-20}$	$2.07\times 10^{-19}$
HEPATOCYTE	362/2316	$2.82\times 10^{-20}$	$3.39\times 10^{-19}$
OMENTUM	350/2316	$3.25\times 10^{-17}$	$3.51\times 10^{-16}$

**Table 3 genes-13-01097-t003:** Top 10 terms in the “Mouse Gene Atlas” category in Enrichr.

Term	Overlap	*p*-Value	Adjusted *p*-Value
mammary gland non-lactating	116/201	$7.92\times 10^{-64}$	$7.61\times 10^{-62}$
skeletal muscle	229/710	$5.23\times 10^{-63}$	$2.51\times 10^{-61}$
liver	243/928	$3.58\times 10^{-48}$	$1.14\times 10^{-46}$
adipose brown	148/456	$5.78\times 10^{-41}$	$1.39\times 10^{-39}$
heart	154/568	$2.53\times 10^{-32}$	$4.86\times 10^{-31}$
kidney	80/554	$3.98\times 10^{-4}$	$5.90\times 10^{-3}$
osteoblast day 21	44/264	$4.30\times 10^{-4}$	$5.90\times 10^{-3}$
bladder	33/195	$1.63\times 10^{-3}$	$1.96\times 10^{-2}$
adipose white	33/199	$2.29\times 10^{-3}$	$2.44\times 10^{-2}$
MEF	45/300	$3.33\times 10^{-3}$	$3.20\times 10^{-2}$

**Table 4 genes-13-01097-t004:** Number of probes selected by other methods.

Tissue	*t* Test	Sam	Limma
Adipose	556	773	116
Muscle	100	119	2
liver	947	1090	211
ComBat	4009	180	0

**Table 5 genes-13-01097-t005:** Top three terms by other methods in the “Mouse Gene Atlas” category in Enrichr.

Term	Overlap	*p*-Value	Adjusted *p*-Value
*t* test			
Adipose			
adipose brown	38/456	$4.17\times 10^{-12}$	$3.92\times 10^{-10}$
mammary gland lact	12/104	$3.87\times 10^{-6}$	$1.82\times 10^{-4}$
macrophage peri LPS thio 0 h	18/353	$1.14\times 10^{-3}$	$3.59\times 10^{-2}$
Muscle			
adipose brown	29/456	$4.34\times 10^{-26}$	$2.26\times 10^{-24}$
heart	21/568	$3.88\times 10^{-14}$	$1.01\times 10^{-12}$
mammary gland lact	4/104	$1.15\times 10^{-3}$	$1.99\times 10^{-2}$
Liver			
liver	90/928	$2.53\times 10^{-16}$	$2.38\times 10^{-14}$
adipose brown	40/456	$9.53\times 10^{-7}$	$4.48\times 10^{-5}$
kidney	40/554	$9.11\times 10^{-5}$	$2.86\times 10^{-3}$
ComBat			
bone marrow	107/413	$1.04\times 10^{-10}$	$9.98\times 10^{-9}$
osteoblast day 21	75/264	$8.31\times 10^{-10}$	$3.99\times 10^{-8}$
embryonic stem line V26 2 p16	149/728	$9.44\times 10^{-7}$	$3.02\times 10^{-5}$
sam			
Adipose			
adipose brown	51/456	$2.61\times 10^{-16}$	$2.48\times 10^{-14}$
mammary gland lact	12/104	$5.67\times 10^{-5}$	$2.69\times 10^{-3}$
macrophage peri LPS thio 0 h	23/353	$3.54\times 10^{-4}$	$1.12\times 10^{-2}$
Muscle			
adipose brown	33/456	$2.16\times 10^{-29}$	$1.21\times 10^{-27}$
heart	23/568	$7.01\times 10^{-15}$	$1.96\times 10^{-13}$
mammary gland lact	4/104	$1.91\times 10^{-3}$	$3.47\times 10^{-2}$
Liver			
liver	93/928	$3.10\times 10^{-14}$	$2.91\times 10^{-12}$
adipose brown	43/456	$1.63\times 10^{-6}$	$7.66\times 10^{-5}$
kidney	43/554	$1.76\times 10^{-4}$	$5.51\times 10^{-3}$
Cell cycle	11/124	$2.95\times 10^{-9}$	$4.71\times 10^{-7}$
Oocyte meiosis	9/129	$6.65\times 10^{-7}$	$5.32\times 10^{-5}$
Progesterone-mediated oocyte maturation	8/100	$1.00\times 10^{-6}$	$5.34\times 10^{-5}$
limma			
Adipose			
adipose brown	14/456	$4.19\times 10^{-8}$	$2.85\times 10^{-6}$
adipose white	4/199	$1.61\times 10^{-2}$	$5.46\times 10^{-1}$
intestine small	6/466	$2.59\times 10^{-2}$	$5.87\times 10^{-1}$
Liver			
liver	33/928	$2.39\times 10^{-11}$	$1.60\times 10^{-9}$
adipose brown	7/456	$1.31\times 10^{-1}$	$1.00\times 10^{0}$
heart	8/568	$1.57\times 10^{-1}$	$1.00\times 10^{0}$

## Data Availability

All of the data used in this study are available in GEO ID GSE13268, GSE13269, and GSE13270.

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
