# Peer review of "Integrated Analysis of Tissue-Specific Gene Expression in Diabetes by Tensor Decomposition Can Identify Possible Associated Diseases"

_genes, 2022, doi:10.3390/genes13061097_

Round 1

Reviewer 1 Report

Dear authors,

I think the title is long. To attract readers, I recommend restructuring it to a more concise one.
There is a minor typing mistake in line 160.

Additionally, the table and figure captions should present more details. For instance, this is the caption of table 1: “Table 1. G(2, 1, 2, 1, 1, â„“6)”. Could you describe it better? Also, could you explain better figure 1's caption?

Furthermore, I think you should have given more focus on the results and the implication for the problem solution. 

Reviewer 2 Report

In the present paper, the authors propose, an integrated analysis by applying a previously proposed tensor decomposition (TD)-based unsupervised feature extraction (FE) method with standard deviation (SD) optimization to microarray-measured gene expression data for diabetes mellitus from multiple tissues. The authors also state that "..found that applying the TD-based unsupervised FE with SD optimization to gene expression profiles from individual tissues can identify diseases associated with diabetes that cannot be identified by the other state-of-the-art methods". The present work is interesting in its conception, yet several issues should be addressed.

In lines 14-15 the authors state "...are singular value matrices and orthogonal matrices". Please explain, why are the described tensors orthogonal.

In lines 32-33 it states "Each of these profiles is composed of gene expression profiles from five individuals seen at two hospitals, and they include data for three tissues" The sentence is very confusing. GEO datasets concern the species Rattus Norvegicus. How is a hospital and individuals involved?

How are the differentially expressed genes derived? the p-value adjustment is the method, and how is the amount of data series taken into account? How was ttest applied to three sets of data?

Figure 1, makes no sense. What does it show, what are the lines and what do they represent? the figure should be better explained.

In addition, the gene annotation for pathway enrichment makes also no sense since there is no previous information on how the DE genes were derived.

Overall, my general comment to the authors is that their manuscript is very difficult to understand and follow. The use of tensors in gene expression surely looks fancy and interesting, yet it pointless if a reader, especially from the biological discipline has no use of it, because it is impossible to understand. I am myself aware of tensors and its uses as well as their mathematical formulation, yet it was impossible to follow the paper.

Some background should have been given to help the reader connect the dots; why use tensors, what is the benefit and how can they be applied to gene expression data?

Finally, the least the authors should do is compare their results with more "classical" approaches and compare the efficiency of their algorithm. There is a plethora of methods for the comparison of microarray data and one would suffice to prove their concept.

Round 2

Reviewer 1 Report

Dear editor and authors,

The authors included some new sentences and attended to some of my concerns. However,  the paper is still challenging to understand. For instance, they included several equations in section 2.2 explaining HOSVD. However, I miss something like a flowchart reporting an overview of their whole pipeline. Furthermore, even with the captions improvement, I cannot understand the importance of figure 1. 

I strongly recommend moving part of these methodological details to supplementary material and giving more highlights to your results and discussions. 

Reviewer 2 Report

The authors have  improved their manuscript. I have also seen other changes probably according to other review comments, with which I agree. Although, I suggest for the manuscript to be accepted, my suggestion to the authors is to make future works more comprehensible for multiple audiences. Complexity is not always good in a scientific work. Especially, for journals such as Genes, whose audience comes from a variety of disciplines including from basic biological sciences.